# Variations of intracellular density during the cell cycle arise from tip-growth regulation in fission yeast

Pascal D Odermatt[1,2†], Teemu P Miettinen[3,4], Joël Lemière[1], Joon Ho Kang[3,5,6], Emrah Bostan[7], Scott R Manalis[3,8,9], Kerwyn Casey Huang[2,10,11]*, Fred Chang[1]*

[1]Department of Cell and Tissue Biology, University of California, San Francisco, San Francisco, United States; [2]Department of Bioengineering, Stanford University, Stanford, United States; [3]Koch Institute for Integrative Cancer Research, Massachusetts Institute of Technology, Cambridge, United States; [4]MRC Laboratory for Molecular Cell Biology, University College, London, United Kingdom; [5]Department of Physics, Massachusetts Institute of Technology, Cambridge, United States; [6]Brain Science Institute, Korea Institute of Science and Technology, Seoul, Republic of Korea; [7]Informatics Institute, University of Amsterdam, Amsterdamn, Netherlands; [8]Department of Biological Engineering, Massachusetts Institute of Technology, Cambridge, United States; [9]Department of Mechanical Engineering, Massachusetts Institute of Technology, Cambridge, United States; [10]Department of Microbiology and Immunology, Stanford University School of Medicine, Stanford, United States; [11]Chan Zuckerberg Biohub, San Francisco, United States

*For correspondence:
kchuang@stanford.edu (KCH);
fred.chang@ucsf.edu (FC)

Present address: [†]Global Health Institute, Swiss Federal Institute of Technology Lausanne, Lausanne, Switzerland

Competing interests: The authors declare that no competing interests exist.

**Abstract** Intracellular density impacts the physical nature of the cytoplasm and can globally affect cellular processes, yet density regulation remains poorly understood. Here, using a new quantitative phase imaging method, we determined that dry-mass density in fission yeast is maintained in a narrow distribution and exhibits homeostatic behavior. However, density varied during the cell cycle, decreasing during G2, increasing in mitosis and cytokinesis, and dropping rapidly at cell birth. These density variations were explained by a constant rate of biomass synthesis, coupled to slowdown of volume growth during cell division and rapid expansion post-cytokinesis. Arrest at specific cell-cycle stages exacerbated density changes. Spatially heterogeneous patterns of density suggested links between density regulation, tip growth, and intracellular osmotic pressure. Our results demonstrate that systematic density variations during the cell cycle are predominantly due to modulation of volume expansion, and reveal functional consequences of density gradients and cell-cycle arrests.

## Introduction

Intracellular density, a cumulative measure of the concentrations of all cellular components, is an important parameter that globally affects cellular function: density affects the concentration and activities of biomolecules and can impact biophysical properties of the cytoplasm such as macromolecular crowding, diffusion, mechanical stiffness, and phase transitions (*Mitchison, 2019*; *Neurohr et al., 2019*; *van den Berg et al., 2017*; *Zhou et al., 2008*). Although it is typically assumed that density must be maintained at a particular level to optimize fitness, there is a growing appreciation that intracellular density often varies across physiological conditions. For example, substantial shifts in density and/or crowding have been detected in development, aging, and disease states (*Neurohr and Amon, 2020*; *Oh et al., 2019*). Even normal cell-cycle progression

can involve changes in density; in cultured mammalian cells, volume increases by 10–30% during mitosis (*Son et al., 2015*; *Zlotek-Zlotkiewicz et al., 2015*), which likely dilutes the cytoplasm prior to an increase in density during cytokinesis.

Despite this biological centrality, the homeostatic mechanisms maintaining cellular density remain poorly understood. Over the course of a typical cell cycle, cells both double their volume and duplicate all cellular contents. A critical unresolved question is how this growth in cell volume (increase in cell size) and biosynthesis are coordinated. In walled cells, the rate of volume growth is dictated largely by cell-wall synthesis and turgor pressure (*Rojas and Huang, 2018*). In principle, feedback mechanisms could tightly couple biosynthesis with wall expansion. However, recent studies have demonstrated that it is possible to decouple biosynthesis from volume growth. For instance, budding yeast cells that are arrested in G1 phase grow to very large sizes and exhibit dilution of the cytoplasm accompanied by decreased protein synthesis and growth rate (*Neurohr et al., 2019*). Conversely, temporary inhibition of volume growth using osmotic oscillations or inhibition of secretion leads to increased density accompanied by a subsequent dramatic increase in volume growth rate in fission yeast (*Knapp et al., 2019*).

Numerous methods have been developed to measure aspects of biomass and intracellular density and crowding in living cells (*Mitchison, 2019*; *Neurohr and Amon, 2020*; *Zangle and Teitell, 2014*). Suspended microchannel resonators (SMRs) infer buoyant cell mass from changes in the frequency of a resonating cantilever as a single cell passes through an embedded microchannel (*Burg et al., 2007*). As an alternative to SMRs, quantitative phase imaging (QPI) is a well-established optical technique for extracting dry-mass measurements from changes in the refractive index (*Park et al., 2018*), as the refractive indices of major cellular components such as proteins, lipids, and nucleic acids are similar (*Zangle and Teitell, 2014*). Previous studies have used phase gratings or holography to generate phase-shift maps that can be used to quantify intracellular density; these approaches require specialized equipment (*Lee et al., 2013*). Precise measurement of density throughout the cell cycle requires non-invasive, long-term quantification of single cells at high spatial and temporal resolution.

Here, we developed a new QPI method for measuring the intracellular density of fission yeast cells. This label-free method is based on the analysis of *z*-stacks of bright-field images (*Bostan et al., 2016*), thus having the advantage of not requiring a specialized phase objective or holographic system. The fission yeast *Schizosaccharomyces pombe* is a leading cell-cycle model, as these rod-shaped cells have a highly regular shape, size, and cell cycle conducive to quantitative analyses (*Hoffman et al., 2015*). Using QPI to track the density of individual cells as they grew and divided within a microfluidic chamber, we determined that wild-type fission yeast cells exhibit characteristic density changes during the cell cycle, in which density falls during G2 phase and increases during mitosis and cytokinesis. These density variations arise from differences in the relative rates of volume and mass growth; while mass grows exponentially throughout the cell cycle, volume growth varies dependent on cell cycle stage/phase. Perturbations to cell-cycle progression and/or growth exacerbated density changes. We further observed gradients in density within single cells and found that intracellular density variations were correlated with tip growth and intracellular pressure. Our findings illustrate a general mechanism by which density is regulated through controlling the relative rates of volume growth and biosynthesis.

## Results

### QPI enables high-resolution measurements of intracellular density in growing cells

To measure intracellular density using a standard wide-field microscope, we developed a version of QPI in which the phase shift is retrieved computationally from a *z*-stack of bright-field images (*Bostan et al., 2016*). This label-free approach takes advantage of the relationship between the intracellular concentration of biomolecules and the refractive index of the cell interior (*Zangle and Teitell, 2014*), which can be computed from the light intensity profile along the *z*-direction using the transport-of-intensity equation (*Figure 1A*, Materials and methods). To calibrate phase shifts with absolute concentrations (dry-mass/volume), we measured the phase shifts within cells grown in media containing a range of concentrations of a calibration standard (bovine serum albumin, BSA)

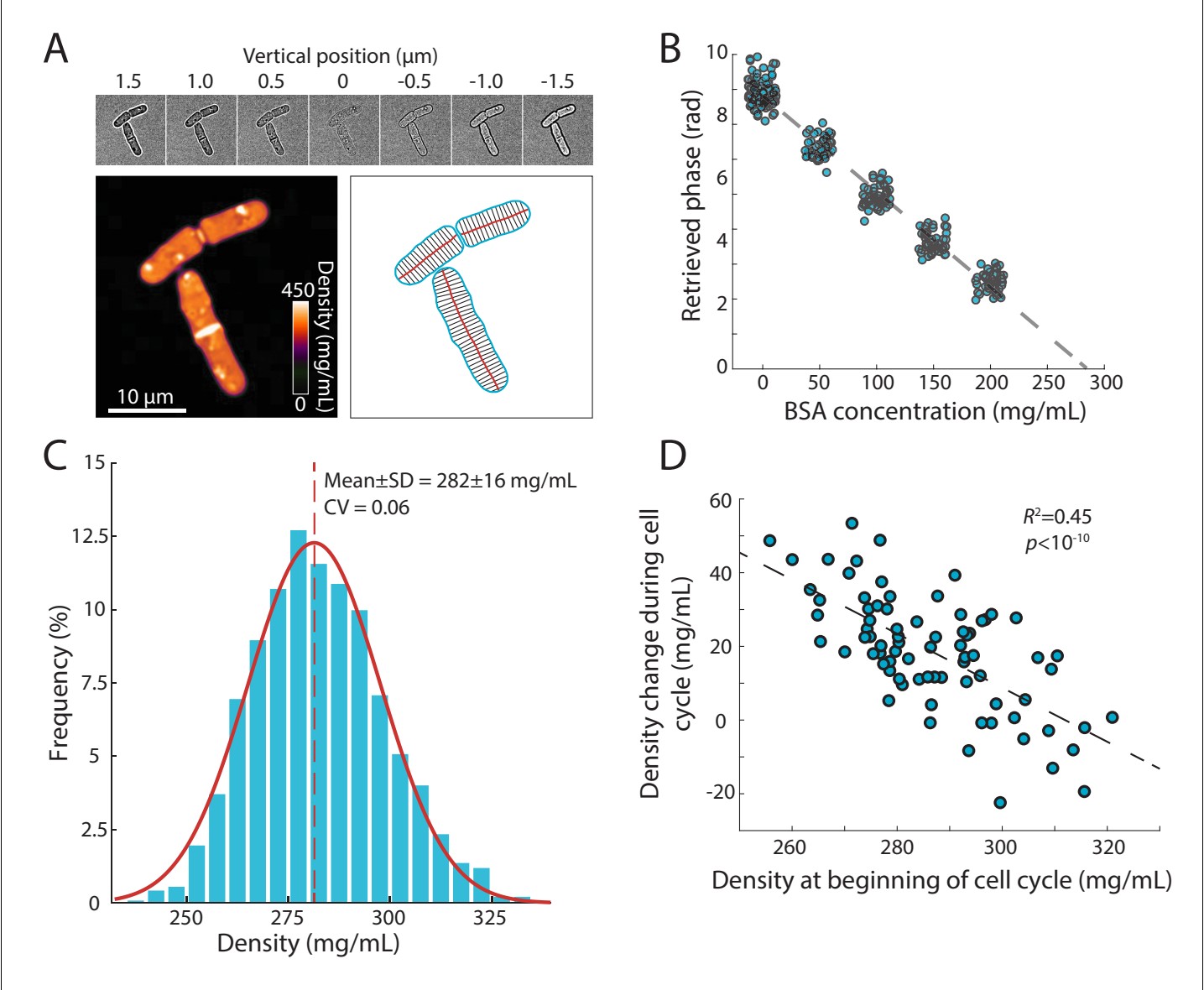

**Figure 1.** Precise measurement of intracellular density using QPI based on a *z*-stack of bright-field images. (**A**) QPI method for computing cytoplasmic density from bright-field images. A *z*-stack of bright-field images of fission yeast cells ±1.5 μm around the mid-plane focal position (top panel) were computationally analyzed by solving the transport-of-intensity equation (*Bostan et al., 2016*) to retrieve pixel-by-pixel phase-shift maps (bottom left). Cellular dimensions were determined via segmentation and skeletonization (bottom right). (**B**) QPI phase shifts were calibrated by imaging cells in media supplemented with a range of concentrations of BSA. The retrieved phase shift is linearly related to concentration (dashed line is the linear best fit). (**C**) Histogram of dry-mass density measurements of exponential-phase fission yeast cells grown at 30°C in YE5S medium. A Gaussian fit (red) yielded a mean (dashed line) density of 282 ± 16 mg/mL (*n* = 2345 time points, 78 cells). SD, standard deviation. (**D**) Cell density at the beginning of the cell cycle was inversely correlated to the change in density during the cell cycle, indicative of homeostatic behavior (*n* = 76 cells). The distribution of density changes was not centered around 0 mg/mL due to a ~ 15 mg/mL density decrease after cell separation (see *Figure 2F*).

The online version of this article includes the following source data and figure supplement(s) for figure 1:

**Source data 1.** Source Data for *Figure 1B* on BSA calibration for QPI.
**Source data 2.** Source Data for *Figure 1C* on density distribution in wildtype cells.
**Source data 3.** Source Data for *Figure 1D* on density homeostasis behavior.
**Figure supplement 1.** Density distribution is robustly maintained.
**Figure supplement 1—source data 1.** Source Data for *Figure 1—figure supplement 1C* on temperature effects on density.

(Materials and methods); these measurements showed a linear relationship that can be used to extrapolate the intracellular density of cells (*Figure 1B*). This method provides pixel-scale measurements of density in living cells and can easily be applied during time-lapse imaging with sub-minute time resolution on most wide-field microscopes.

Using this methodology, we determined that the mean dry-mass density of an asynchronous population of wild-type *S. pombe* cells growing at 30°C in rich YE5S medium was 282 ± 16 mg/mL (*Figure 1C*). There were various sources of intracellular heterogeneities such as lipid droplets (*Figure 1—figure supplement 1A*) and cell-wall septa, both of which were regions of high signal (*Figure 1A*). The nucleus was not distinguishable in most phase-shift maps, indicating similar density as the cytoplasm (*Figure 1A*). Nonetheless, the distribution of densities was remarkably narrow, with a coefficient of variation (CV) of 0.06, despite variability in cell size and cell-cycle stage. For cells within the same cell-cycle stage, the distribution of densities was even narrower (CV <0.05, *Figure 1—figure supplement 1B*). To determine whether cells could recover from fluctuations in density, we examined the change in density over a cell cycle as a function of initial density at the beginning of the cell cycle. These two quantities were inversely related (*Figure 1D*), indicating a homeostatic behavior that maintains density. A similar distribution of densities was observed in cells grown at 25°C and after temperature shifts (*Figure 1—figure supplement 1C*). Together, these results suggest that dry-mass density is robustly maintained, and demonstrate that this QPI approach can precisely measure absolute dry-mass density in living cells with high temporal and spatial resolution.

## Intracellular density follows a characteristic trajectory during the *S. pombe* cell cycle

To determine whether intracellular density changes over the course of the fission yeast cell cycle, we imaged proliferating cells in time-lapse using QPI in a microfluidic device under constant flow of growth medium (Materials and methods; *Figure 2A*, *Figure 2—video 1*). Density maps were segmented to extract cellular dimensions, from which volume was computed (Materials and methods; *Figure 1A*). Total dry mass of each cell was computed from volume and mean density measurements. We imaged cells throughout their entire cell cycle, and then aligned the computed data from each cell by relative cell-cycle progression, from cell birth (first detectable physical separation between daughter cells) until just before cell-cell separation at the end of the cell cycle.

Intracellular density displayed consistent dynamics during the *S. pombe* cell cycle. As observed previously (*Mitchison and Nurse, 1985*), volume measurements showed that cells exhibited steady tip growth in interphase (mostly G2 phase), and then volume growth slowed or halted during mitosis and cytokinesis (defined here as the period starting from septum formation and ending at daughter-cell separation; *Figure 2B*). Density gradually decreased from the beginning of the cell cycle through G2 phase by ~5%, followed by a steady rise during mitosis and cytokinesis (*Figure 2C*, *Figure 2—figure supplement 1A*). By contrast to volume, dry mass increased steadily throughout the cell cycle without any obvious transitions (*Figure 2D*). These measurements suggest that the density increase during mitosis and cytokinesis is a consequence of continued mass accumulation when volume growth halts.

We confirmed these density variations using complementary approaches. We used a suspended microchannel resonator (SMR) to measure the buoyant mass of single cells in two different density solutions and thereby derive single-cell buoyant densities and volumes (*Grover et al., 2011*; Materials and methods). SMR measurements revealed that fission yeast exhibit a mean buoyant density of 1.108 ± 0.005 g/mL, and buoyant densities were lower in intermediate-sized cells than in small and large cells (*Figure 2E*, *Figure 2—figure supplement 1B,C*), consistent with our QPI-based observations of cell cycle-dependent densities (*Figure 2C*, *Figure 2—figure supplement 1A*). As expected, the relative changes in buoyant density were much less than changes in dry-mass density due to the different nature of the respective measurements: dry-mass density accounts only for a fraction of buoyant density as cells are comprised mostly of water, and hence relative changes are magnified (*Feijó Delgado et al., 2013*). The general cell-cycle variations in density were also confirmed using holographic methods (*Figure 2—figure supplement 1D,E*; *Rappaz et al., 2009*).

We also identified an additional change in density at cell birth. At the end of cytokinesis, the middle layer of the septum is digested, and daughter cells separate (*Martín-Cuadrado et al., 2003*; *Hercyk et al., 2019*). During this 5–10 min period, the septum bulges outward on each side to form

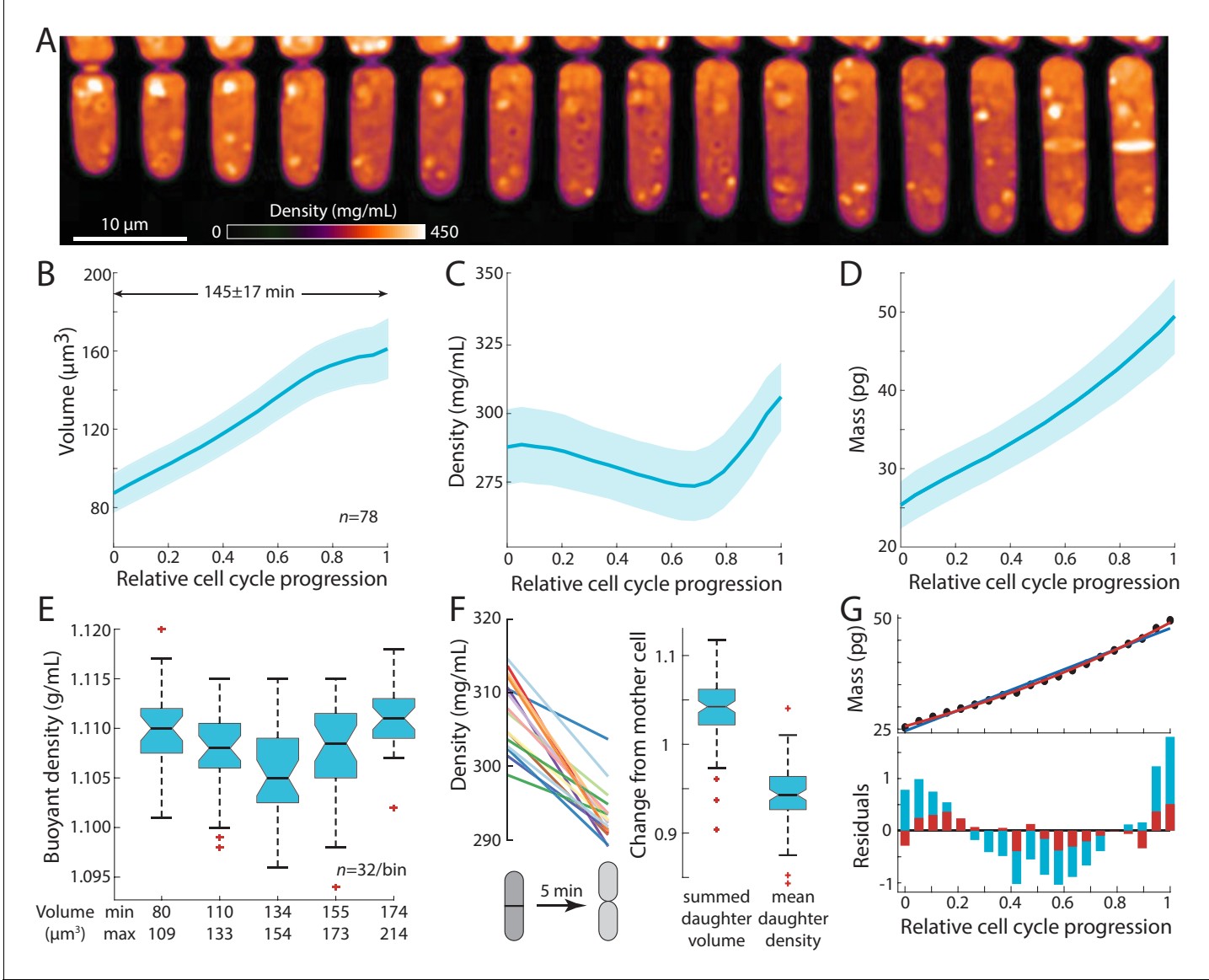

**Figure 2.** Intracellular density varies across the cell cycle. (A) Wild-type fission yeast cells in exponential phase were imaged in time lapse in a microfluidic chamber and phase-shift maps were extracted by QPI. Shown are images of a representative cell traversing the cell cycle from (left) cell birth to (right) septation (10 min/frame). (B,C,D) Cell volume (B), density (C), and dry mass (D) of cells aligned by their relative progression in the cell cycle. Curves are mean values and shaded regions represent one standard deviation (n = 78 cells). Mass was estimated from volume and density measurements. (E) SMR-based measurements of fission yeast cells in an asynchronous culture (binned by cell volume) showed a similar decrease in buoyant density at intermediate volume as QPI measurements of dry-mass density (C). (F) Cell density decreases upon cell separation. Left: density measurements of cells just before and just after cell separation (5 min apart). Right: normalized changes in volume and density between the mother and resultant daughter cells. (G) Dry mass grows more exponentially than linearly. The residuals (bottom) of an exponential fit (red) to mass growth (D) were much smaller than a linear fit (blue). See also *Figure 2—figure supplement 3A*.

The online version of this article includes the following video, source data, and figure supplement(s) for figure 2:

**Source data 1.** Source Data for *Figure 2* on density measurements during the cell cycle.

**Figure supplement 1.** Measurements of buoyant density and buoyant mass using SMR and refractive index measurements using holography and QPI show cell-cycle-dependent density variations.

**Figure supplement 1—source data 1.** Source Data for *Figure 2E* and *Figure 2—figure supplement 1* on buoyant density measurements.

**Figure supplement 2.** The mean density of *S. pombe* daughter cells was typically lower than the density of the mother cell.

**Figure supplement 2—source data 1.** Source Data for *Figure 2F* and *Figure 2—figure supplement 2* on mother/daughter density differences.

**Figure supplement 3.** Cellular surface area-to-mass ratio varies less than dry-mass density during the *S. pombe* cell cycle.

**Figure supplement 3—source data 1.** Source Data for *Figure 2—figure supplement 3B* on septum area calculation.

*Figure 2 continued on next page*

*Figure 2 continued*

**Figure supplement 3—source data 2.** Source Data for *Figure 2* and *Figure 2—figure supplement 3* on volume, mass, and area relationships.
**Figure 2—video 1.** QPI density maps of wild-type fission yeast cells.
https://elifesciences.org/articles/64901#fig2video1

the rounded shape of the new end, in a physical process driven by turgor pressure (*Atilgan et al., 2015*). QPI analysis of individual cells showed a consistent drop in density within a 5-min window around cell separation (*Figure 2F*, *Figure 2—figure supplement 2*); during this period, cell volume increased by ~5% while density decreased by ~5% (*Figure 2F*). The magnitude and rapidity of the

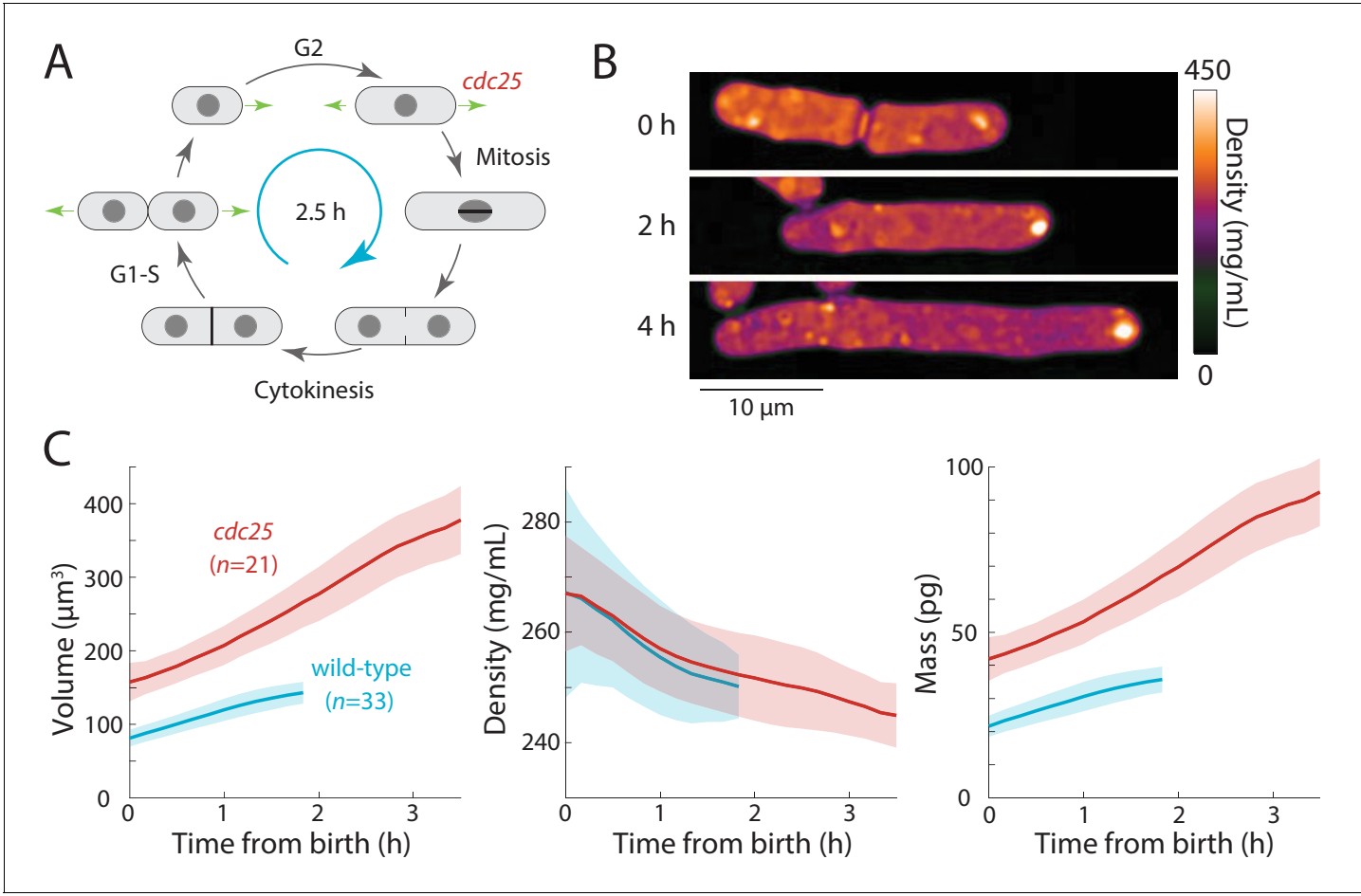

**Figure 3.** Extension of the G2 phase of the *S. pombe* cell cycle results in cell elongation and decreased intracellular density. (A) Schematic of the fission yeast cell cycle, highlighting the point at which the *cdc25* temperature-sensitive mutant delays the G2-M transition. (B) *cdc25-22* cells were shifted from the permissive temperature 25°C to the semi-permissive temperature 32°C to extend G2 phase, leading to continued cell elongation. QPI density maps of a representative cell are shown. Note that density decreased during cell elongation. (C) QPI-based measurements of volume (left), density (middle), and dry mass (right) of *cdc25-22* cells (red) that grew at least 2.5-fold relative to their birth length before dividing, compared with wild-type cells (blue) under the same conditions. Measurements are aligned from cell birth until elongation rate decreased to 20 nm/min (as an indication of the transition to mitosis). Curves are mean values and shaded regions represent one standard deviation.
The online version of this article includes the following source data for figure 3:

**Source data 1.** Source Data for *Figure 3* on cdc25 mutant experiment.

density change suggest that the volume increase at cell separation is due primarily to swelling from water uptake. Thus, density appears to be directly tied to the dynamics of volume expansion.

Our precision measurements of cellular dimensions and intracellular density provide a quantitative characterization of dry-mass dynamics (biosynthesis) throughout the cell cycle. The absolute rate of dry-mass accumulation steadily increased during the cell cycle (*Figure 2D*). Dry-mass dynamics were more exponential than linear in nature (*Figure 2G*, *Figure 2—figure supplement 3A*), as shown by a comparison of linear versus exponential fits (*Figure 2G*) and in the dynamics of normalized mass growth (*Figure 2—figure supplement 3A*). The absolute rate of mass synthesis was therefore higher during mitosis and cytokinesis than at the beginning of the cell cycle, even though the cell slowed in volume growth late in the cycle. Thus, mass production was not tightly coupled to volume growth. These measurements suggest a simple model in which the increase of density in mitosis and cytokinesis arises as consequence of continued mass accumulation when volume growth is halted.

## Cell-cycle perturbations exacerbate cell cycle-dependent density variation

Our data demonstrate that intracellular density increases during mitosis and cytokinesis as a result of biosynthesis continuing unabated while volume growth slows; conversely, density decreases during interphase because the rate of volume growth surpasses the rate of mass synthesis. However, the origin of these dynamics remains unresolved. One possibility is that the relative rates of mass synthesis and volume growth are not directly coupled; with mass growing exponentially, density variations then arise indirectly as a consequence of cell-cycle regulation of volume expansion, which is controlled by cell polarity programs that redirect the cell wall growth machinery to the middle of the cell for septum formation prior to cytokinesis (*Simanis, 2015*; *Ray et al., 2010*; *Martin and Arkowitz, 2014*). Alternatively, the density at each cell-cycle stage could be directly programmed to specific levels by specific cell-cycle regulators. Another possibility is that density variations may be due to a cell cycle-independent oscillator, such as a metabolic oscillator (*Papagiannakis et al., 2017*; *Liu et al., 2020*). To distinguish these models, we examined the consequences of arresting or delaying cells at particular stages of the cell cycle (*Figure 3A*). If there is no strict control of biosynthesis then when mitosis or cytokinesis is blocked, mass should continue to accumulate and density should reach higher levels than in normal cells, and conversely density should fall below normal in extended interphase. If density is instead regulated at specific levels according to cell-cycle phase, density levels should not change during cell-cycle delays beyond the ranges appropriate for each phase. If a cell cycle-independent oscillator governs density variations, then oscillations could continue even during cell-cycle arrests (*Novak and Mitchison, 1986*; *Mitchison, 2003*).

First, we tested whether density would decrease further in *S. pombe* cells experiencing an extended period of growth during interphase. We delayed cells harboring a temperature-sensitive *cdc25-22* mutation (*Nurse et al., 1976*) in G2 phase by shifting them from room temperature to the semi-permissive temperature (32˚C). These cells continued to grow from their tips and formed abnormally elongated cells (*Figure 3B,C*) before dividing. To focus on cells that remained in G2 phase for an extended interval, we limited our analysis to cells that elongated to >2.5 fold their initial length. In these mutant cells, during their prolonged G2 phase of 2–3 hr, intracellular density decreased further than in wild-type cells (~8% in *cdc25-22* cells from 267 ± 19 to 245 ± 6 mg/mL, compared to ~5% in wild-type cells from 267 ± 11 to 250 ± 6 mg/mL) (*Figure 3C*). No evidence of density oscillations in the prolonged G2 phase was evident, arguing against cell-cycle independent oscillations as the cause of density variations during a normal cell cycle. These data suggest that density falls during G2 phase because the rate of volume growth continues to be slightly faster than the rate of biosynthesis.

Second, we tested whether cells arrested in mitosis displayed increased intracellular density (*Figure 4A*). We delayed cells in metaphase using a temperature-sensitive *cut7-446* mutant (kinesin-5) defective in mitotic spindle assembly (*Hagan and Yanagida, 1990*). Using QPI, we tracked intracellular density from mitosis initiation until the earliest signs of septum formation. As expected, at the non-permissive temperature this interval was longer for *cut7-446* cells (20–30 min) than for wild-type cells (10–20 min; *Figure 4B,C*). During this mitotic period, the density of wild-type and *cut7-446* cells increased at a similar rate (*Figure 4B,C*): hence, the extended time in metaphase in *cut7-446* cells led to a greater density increase (7% in *cut7-446* from 270 ± 12 to 288 ± 14 mg/mL versus 5% in wild-type cells from 264 ± 11 to 278 ± 13 mg/mL).

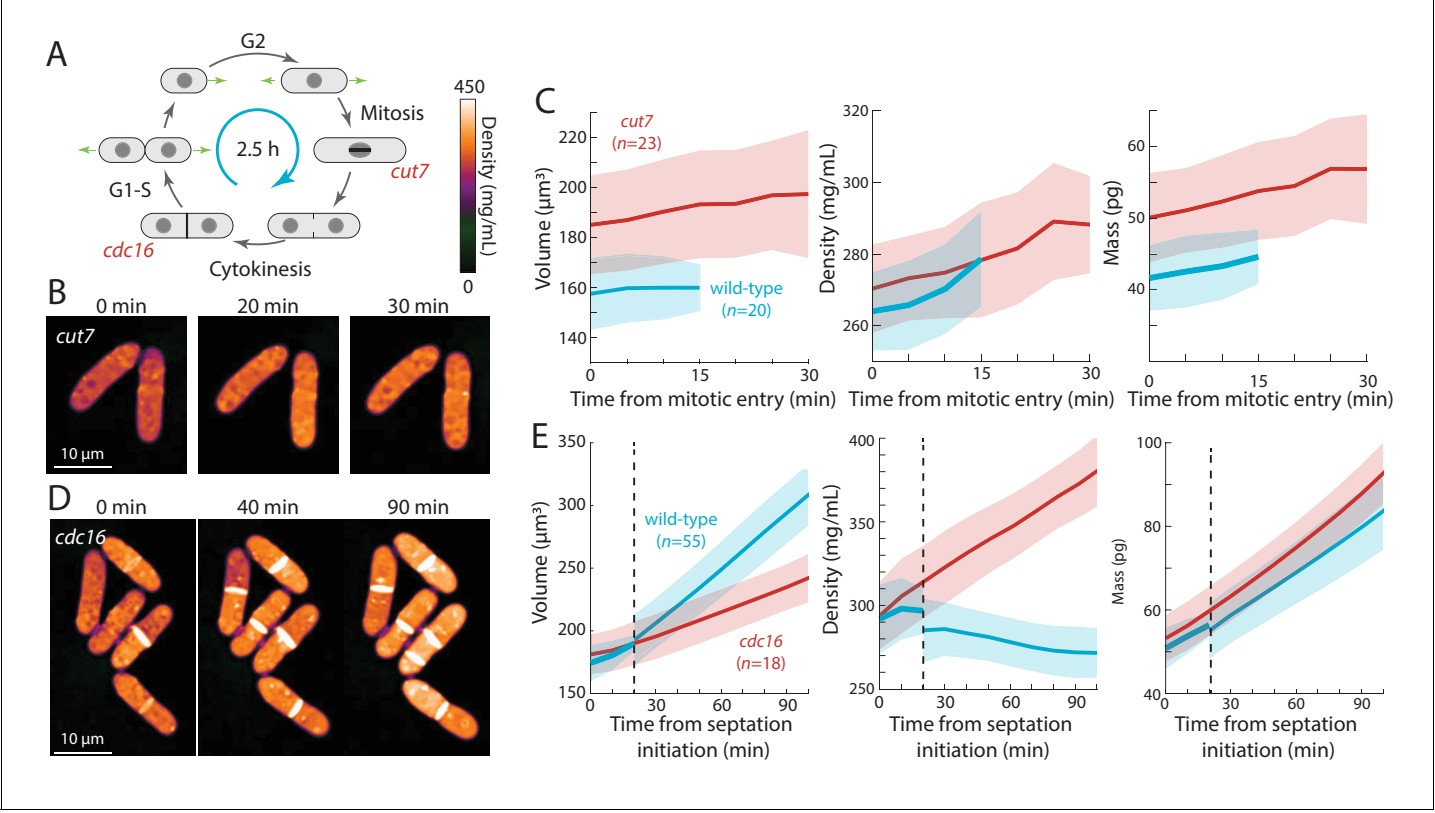

**Figure 4.** Cell-cycle arrests in mitosis and cytokinesis result in increased intracellular density in *S. pombe*. (A) Temperature-sensitive mutants *cut7-446* (spindle kinesin-5) and *cdc16-116* block the cell cycle in mitosis and cytokinesis, respectively. (B) *cut7-446* cells were shifted from 25°C to 30°C to delay mitotic progression. QPI of two representative cells delayed in mitosis for ~20 min until the onset of septation (30 min time point). Note that density continued to increase during mitotic arrest. (C) QPI-based measurements of volume (left), density (middle), and dry mass (right) of *cut7-446* cells from mitotic entry ($t = 0$) through initiation of septum formation at cytokinesis. Curves are mean values and shaded regions represent one standard deviation. (D) *cdc16-116* cells were shifted from 25°C to 34°C to arrest cells in cytokinesis. QPI of five representative cells are shown. *cdc16* cells generally did not complete cell separation and often assembled additional septa without elongating. Note that density increased during this cytokinetic arrest. (E) QPI-based measurements of volume (left), density (middle), and dry mass (right) of *cdc16-116* cells from initiation of the first septum ($t = 0$). Wild-type cells separated after ~20 min (dashed line), and thereafter the behavior of the daughter cells was tracked for comparison with *cdc16* cells (volume and dry mass were summed for the two daughter cells). Curves are mean values and shaded regions represent one standard deviation.

The online version of this article includes the following source data for figure 4:

**Source data 1.** Source Data for *Figure 4* on cdc7 and cdc16 mutant experiment.

Third, we arrested cells in cytokinesis, again to test for an increase in density (*Figure 4A*). *cdc16-116* mutant cells arrest in cytokinesis at the restrictive temperature, and thus repeatedly make septa without elongating (*Minet et al., 1979*). Upon a shift from 25°C to the non-permissive temperature (34°C), cells that maintained cytokinetic arrest continued to increase in cytoplasmic density; density after 90 min was 20–30% higher than in cytokinesis-competent wild-type cells (*Figure 4D,E*). Thus, biosynthesis continues throughout an extended block of mitosis or cytokinesis, leading to abnormally high intracellular density.

Finally, we asked whether inhibition of volume growth is sufficient to increase cytoplasmic density. We previously showed that two treatments that slow volume growth (osmotic oscillations and treatment with brefeldin A) led to an increase in cytoplasmic density (*Knapp et al., 2019*). However, since these treatments do not completely halt volume growth and/or result in cell death, we treated wild-type cells with the F-actin inhibitor latrunculin A, which causes immediate cessation of tip growth

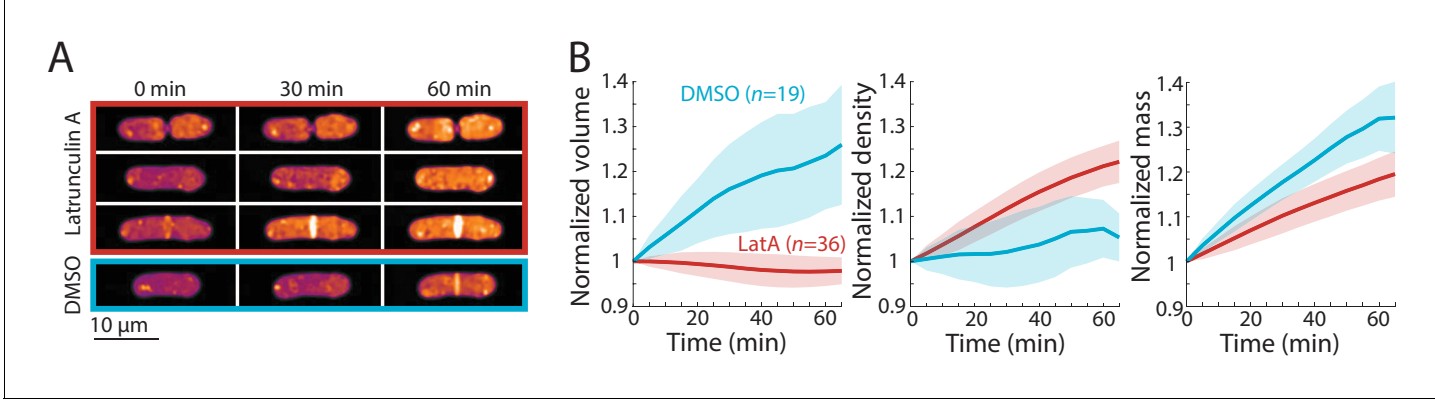

**Figure 5.** Cell-cycle-independent growth inhibition by latrunculin A results in increased intracellular density in *S. pombe*. (**A**) Latrunculin A treatment inhibited cell growth and increased intracellular density regardless of cell-cycle stage. Representative QPI density maps of three wild-type cells at different points of the cell cycle treated with 200 µM latrunculin A for the indicated times. As a control, cells were treated with 1% DMSO; growth continued and density remained relatively constant. (**B**) QPI-based volume (left), density (middle), and dry mass (right) measurements of latrunculin A (LatA)-treated or DMSO-treated wild-type cells from the start of treatment (*t* = 0). Growth halted and density increased due to continued mass synthesis during treatment. Note that the average density in the DMSO control rose slightly due to the removal of cells within the asynchronous population from analysis after cell division. Curves are mean values and shaded regions represent one standard deviation.

The online version of this article includes the following source data and figure supplement(s) for figure 5:

**Source data 1.** Source Data for *Figure 5* on effect of latrunculin A treatment on intracellular density.

**Figure supplement 1.** The increase in intracellular density due to treatment with the actin inhibitor latrunculin A is not dependent on cell size.

**Figure supplement 1—source data 1.** Source Data for *Figure 5—figure supplement 1* on latrunculin A effect on density of cells of different sizes.

independent of cell-cycle stage (*Pan et al., 2014*; *Mutavchiev et al., 2016*). QPI density maps showed that latrunculin A treatment caused all cells to immediately halt tip growth and begin to steadily increase in density, regardless of cell-cycle stage (*Figure 5A,B*). The mean density increase after 1 hr was ~20% (*Figure 5B*). Similar increases in density were seen in cells of different sizes (*Figure 5—figure supplement 1*). However, we noted that in contrast to the mitotic and cytokinesis arrests, mass increases were variable and on average increased more slowly during latrunculin A treatment than during normal growth (*Figure 5B*, *Figure 2D*), suggesting a partial slowdown in biosynthesis and/or an increase in degradation.

Taken together, these results show that cell density differences are exacerbated during cell cycle arrests and that inhibition of volume growth is sufficient to increase intracellular density. These findings are not consistent with models in which density is set at cell-cycle stage-specific levels or involving cell cycle-independent oscillators. Rather, our results strongly support a model in which the variations in intracellular density arise from cell-cycle-dependent changes in volume growth rate.

## A polarized density gradient is associated with the pattern of tip growth

Fission yeast cells have a well-known pattern of volume growth in which after cell division, the old end initiates tip growth soon after cell birth, and partway through G2, tip growth at the new end begins, but at a slower rate than the old end (or not all) (*Mitchison and Nurse, 1985*; *Chang and Martin, 2009*). As expected, our time-lapse data showed that the old and new ends grew over the course of the cycle on average by ~4 and 2 µm, respectively. In QPI density maps, we noted that many cells exhibited a gradient of intracellular density in which the ends that were actively growing appeared less dense than the non-growing ends (*Figure 6A*). We hypothesized that these subcellular gradients reflected differences in tip growth between the two ends of the cell. In agreement with our hypothesis, the slower-growing new end typically appeared denser than the faster-growing end (*Figure 6A*). In some cells, the difference in densities between the fast- and slow-growing ends

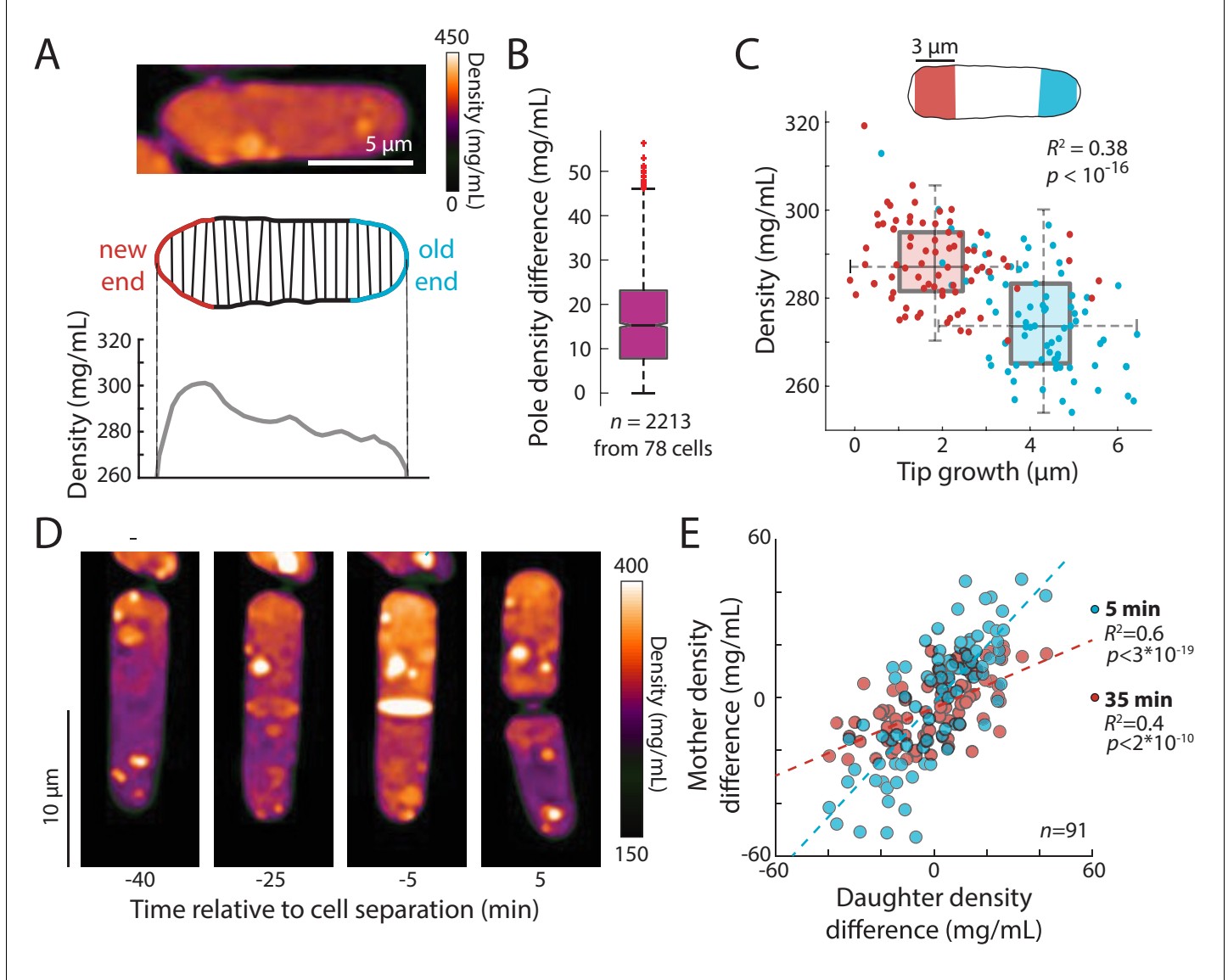

**Figure 6.** An intracellular density gradient negatively correlates with tip growth in *S. pombe*. (**A**) Top: QPI of a representative cell displaying an intracellular gradient of density. Middle: Density was measured in slices perpendicular to the long axis. Bottom: The new end (non- or slowly growing) exhibited a higher density than the old (growing) end. (**B**) Density was substantially different between the new and old ends in many cells. Time-lapse QPI was used to measure the densities in regions within 3 μm of each cell end. Shown is the density difference between the cell poles averaged over the cell cycle. Box extends from 25th to 75th percentile, with the median as a horizontal bar. Whiskers indicate extreme points not considered outliers (*n* = 78 cells). (**C**) Mean density and amount of tip growth over an entire cell cycle was measured in 3 μm regions at the old (blue) and new (red) ends. Old ends grew more and exhibited lower mean densities over the course of the cell cycle than new ends. Box and whiskers plot is as described in (**B**). (**D**) QPI density map of a representative cell at interphase, the start of septum formation, late in septum formation, and after cell division. The gradient of intracellular density in the interphase cell was maintained over time and passed on to the daughter cells. (**E**) Asymmetric density patterns are propagated to the next cell cycle. The density difference between the two ends of a mother cell correlated with the density difference between the progeny daughter cells. Shown are Pearson's correlation coefficient between the density difference of daughter cells and the corresponding halves of the mother cell at 5 min (blue) or 35 min (red) before cell division. The halves of the mother cell exhibited a larger range of density differences at the later time point, when they were more consistent with the density differences between daughter cells.

The online version of this article includes the following source data and figure supplement(s) for figure 6:

**Source data 1.** Source Data for *Figure 6* on density relationship to tip growth.

**Figure supplement 1.** Spatial intracellular density gradients are present in cells with similar widths, and are stably maintained in cells treated with latrunculin A.

**Figure supplement 1—source data 1.** Source Data for *Figure 6—figure supplement 1A* on density gradient.

**Figure supplement 2.** Total protein and RNA exhibit intracellular spatial gradients.

*Figure 6 continued on next page*

*Figure 6 continued*

**Figure supplement 2—source data 1.** Source Data for *Figure 6—figure supplement 2* on FITC staining.

was ~10% of the mean overall density (*Figure 6A*). The mean density difference between the two ends throughout the cell cycle was ~15 mg/mL, corresponding to ~5% of the mean overall density (*Figure 6B*). To address the potential for differences in the widths (and hence heights above the coverslip) of old and new ends to influence phase shifts (*Figure 6—figure supplement 1A*), we constrained our analysis to cells within a narrow range of widths and found that local density and tip growth remained highly correlated (*Figure 6—figure supplement 1B*). Examples of post-cytokinesis cells with adjacent compartments exhibiting different densities (*Figure 7*, *Figure 7—figure supplement 1*) further indicated that these differences could not be explained simply by width differences. We further confirmed the presence of spatial gradients by staining cells for total protein and RNA using the dye fluorescein isothiocyanate (FITC). FITC intensity differed by ~5% between the old and new end of monopolar cells that are growing only from the old end (*Figure 6—figure supplement 2*), consistent with the gradient in QPI density maps. Since tip growth is regulated by actin-dependent mechanisms (*Pan et al., 2014*; *Mutavchiev et al., 2016*; *Chang and Martin, 2009*), we tested whether maintenance of the gradient is dependent on F-actin or tip growth: we found that in cells treated with latrunculin A, spatial density gradients persisted over time (*Figure 6—figure supplement 1C*). These results demonstrate that intracellular density gradients are stable and linked to local growth patterns, and that their maintenance does not require active growth or F-actin.

We noticed asymmetries in density between daughter cells after cell division, in which one of the daughters was denser than its sister. Time-lapse imaging showed that the intracellular density differences established during interphase were often propagated through cell division and correlated with density differences between the progeny daughter cells after cytokinesis (*Figure 6D,E*). Thus, subcellular density variations are stable enough to be propagated through generations.

## Cell density differences are linked to intracellular osmotic pressure

Next, we ascertained whether density variations of 5–20% – the magnitude observed within and across cells with normal physiology – have physiological consequences. One variable potentially connected with intracellular density is macromolecular crowding. High concentrations of macromolecules are predicted to produce colloid osmotic pressure that may influence cell mechanics (*Mitchison, 2019*). As noted above, the densities of daughter compartments in septated cells were often different from each other (*Figure 6D,E*); these differences were often exacerbated in cells with cell-division defects, such as *mid1*, *mid2*, and *cdc16* cells. We noted that the septum between these daughter cells generally bent away from the more dense compartment (*Figure 7A*, *Figure 7—figure supplement 1A*). A similar situation was observed in multi-septated cells in which the internal cellular compartment, whose volume growth is restricted by two septa, increased in density (*Figure 7A*, right). Previous studies (*Atilgan et al., 2015*) reported that the septum is an elastic structure that can be used as a biosensor that reports on osmotic pressure differences between compartments. For instance, when one daughter is lysed by laser microsurgery and loses turgor pressure, the septum bulges away from the intact daughter (*Atilgan et al., 2015*). Temporal fluctuations in septum bending have been proposed to arise from fluctuations in pressure differences between daughter cells (*Muñoz et al., 2013*).

To investigate the relationship between density differences and septum bending, we focused on *mid2Δ* cells. Mid2 is an anillin ortholog that regulates septins in late cytokinesis; *mid2Δ* mutants exhibit long delays (1–2 hr) in cell separation and thus most cells in the population have one or more septa (*Berlin et al., 2003*). We used QPI to track intracellular density over time and identified the time after septum formation at which the maximum density difference was reached for each cell. Of the 71% (64/90) of cells exhibiting a bent septum, 97% (62/64) exhibited a septum bent away from the compartment of higher density at the time of maximum density difference (*Figure 7A*), with a

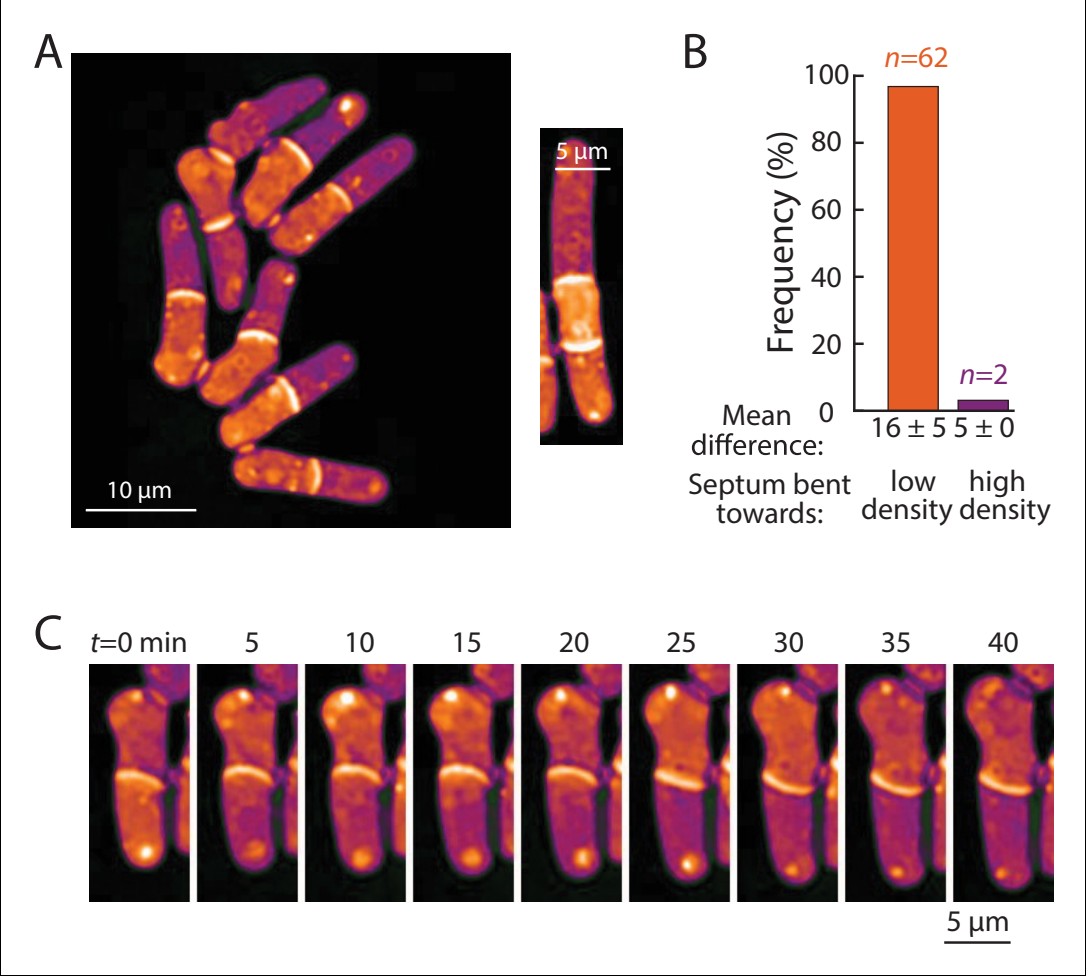

**Figure 7.** Bending of the septum reveals a link between intracellular density and osmotic pressure. (**A**) Left: QPI density map of *mid2Δ cells* that are delayed in cell separation showed bent septa and density differences between two daughter-cell compartments. Note that the septa are bent away from the denser compartment in all cells in this image. Right: In multi-septated *mid2Δ* cells, internal compartments bounded by two septa exhibited higher density than the surrounding compartments; in these situations, both septa typically bent away from the higherdensity compartment. (**B**) The direction of septum bending was almost always toward the lower density daughter cell. The direction of the bent septum was measured at the time of maximum density difference between daughter-cell compartments. (**C**) An example of a cell in which the direction of septal bending and the sign of the density difference between daughter-cell compartments fluctuated over time. After the bottom compartment decreased and the top compartment increased in density, the septum bent in the opposite direction, consistent with the correlation between bending and density difference in (**B**).

The online version of this article includes the following figure supplement(s) for figure 7:

**Figure supplement 1.** Septa bend away from the compartment of higher density in *mid2* and *cdc16* mutant cells.

mean maximum difference of 16 ± 5% (*Figure 7B*). In 2/90 cells, the septum was bent in the opposite manner, toward the compartment of lower density; in these rare instances, the maximum density difference was substantially lower (5%; *Figure 7B*) and may represent cases in which the septum is fluctuating in direction. Indeed, in one cell, the fluctuating direction of septum bending coincided with alternation of the sign of the density difference between the daughter cells (*Figure 7C*). In instances where the septum appeared flat, the density difference was substantially lower (~4.5%)

than in cells with a bent septum. The observation that septal bending occurred for density differences as low as 5–10% suggests that the density variations over the course of a normal cell cycle (*Figure 2*), in cell cycle-arrested cells (*Figures 3*, *4* and *5*) or between growing and non-growing cell tips (*Figure 6*) may reflect substantial changes in turgor-mediated stresses.

## Discussion

Here, we established a QPI method based on *z*-stacks of bright-field images for quantifying intracellular density dynamics in living cells. We found that exponential-phase fission yeast cells have a dry-mass density of 282 ± 16 mg/mL (*Figure 1C–E*) and a buoyant-mass density of 1.108 ± 0.005 g/mL (*Figure 2E*), comparable to measurements in other organisms (*Neurohr and Amon, 2020*; *Bryan et al., 2010*). Density varied systematically across the cell cycle in wild-type fission yeast cells over a range of ~10% (*Figure 2C*), while the relative rate of dry-mass synthesis remained constant (reflecting exponential accumulation) throughout all cell-cycle stages (*Figure 2F,G*). These quantitative findings, which utilize precise sub-pixel measurements of cellular dimensions and automated analysis platforms, are consistent with more qualitative density studies of fission yeast using other methods (*Rappaz et al., 2009*; *Mitchison, 1957*; *Stonyte et al., 2018*).

Our data support a model in which these density variations during the cell cycle are a product of programmed changes in volume growth accompanied by a constant relative rate of mass biosynthesis. Accordingly, volume growth and biosynthesis were not tightly coupled throughout the cell cycle. During tip growth in G2 phase, intracellular density dropped steadily (*Figures 2C* and *3C*), indicating that the rate of volume growth outpaces biosynthesis during this period. Density steadily rose during mitosis and cytokinesis (*Figure 2C*), when volume growth ceases or slows. After separation of daughter cells (cell birth), density dropped during the rapid increase in cell volume as the new cell end expanded (*Figure 2F*; *Atilgan et al., 2015*), possibly due to water influx.

Consistent with this model, density shifts were exacerbated by perturbations of cell-cycle progression or of volume growth directly. *cdc25* mutants delayed in G2 phase exhibited a steady decline in density as cells elongated abnormally (*Figure 3*), reminiscent of the cytoplasmic dilution observed in very enlarged budding yeast cells and senescent mammalian cells (*Neurohr et al., 2019*). During mitotic arrest at the spindle checkpoint (*cut7*) or cytokinesis arrest through regulation of the SIN pathway (*cdc16*), volume growth was slowed but mass synthesis was unaffected, resulting in steady density increases (*Figure 4C,E*). Inhibition of cell growth with latrunculin A caused a steady increase in density regardless of cell-cycle stage (*Figure 5*). These findings suggest that any perturbation that affects cell-cycle progression or growth is likely to alter density dynamics. Cell-cycle arrests are commonly used to synchronize cells and are often triggered in response to stresses such as DNA damage. Our study demonstrates that such perturbations are not as innocuous as often thought; arrests not only affect cell-cycle progression and cell size, but also cause changes in intracellular density and potentially other downstream changes in physiology.

Our studies provide quantitative measurements of the mass dynamics of individual fission yeast cells throughout the cell cycle. We found that mass accumulation was continuous, consistent with previous studies (*Mitchison, 1957*; *Stonyte et al., 2018*). Notably, our quantitative measurements suggested that mass dynamics were exponential in nature (*Figure 2D,G*, *Figure 2—figure supplement 3A*), consistent with findings in other cell types (*Godin et al., 2010*). Intriguingly, studies of density and growth in other cell types reported somewhat different cell-cycle patterns. In budding yeast, buoyant density is lowest in early G1 and rises in late G1 and S phase at the time of bud formation (*Bryan et al., 2010*). Moreover, cell-cycle arrests in S and M phase and latrunculin A treatment do not lead to increases in buoyant density in budding yeast, unlike our findings in fission yeast (*Figure 5*). In human cells, mass growth continues in early mitosis, but stops in metaphase and resumes in late cytokinesis, potentially with subtle oscillations (*Liu et al., 2020*; *Miettinen et al., 2019*). Density is constant in human cells during much of the cell cycle, but decreases in mitosis (by 0.5% in buoyant mass, equivalent to >10% decrease in dry-mass density) coincident with a 10–30% volume increase during mitotic rounding; density then slightly increases in cytokinesis (*Son et al., 2015*; *Zlotek-Zlotkiewicz et al., 2015*). In the bacterium *Escherichia coli*, density varies somewhat from birth to division, but the ratio of surface area to mass is relatively constant, suggesting that biosynthesis is linked to surface-area synthesis (*Oldewurtel et al., 2019*). Here, we found that in fission yeast, mass was also more closely coupled with surface area than volume, especially when the

surface area of both sides of the septum were factored in (*Figure 2—figure supplement 3B–D*). However, examples such as latruculin A-treated cells (*Figure 5B*) demonstrated that mass accumulation and surface area growth are not inextricably linked and can be uncoupled. It remains to be seen whether general rules of cell-density regulation will emerge from comparisons across organisms and cell types.

Our findings also provide insight into the relationship between the rate of volume growth and subcellular density regulation. Fission yeast cells grow through tip growth, which involves the extension and assembly of new cell wall and plasma membrane at the growing tip; this growth is accomplished by a complex integration of the cell-polarity machinery, exocytosis, wall growth and mechanics, and turgor pressure (*Martin and Arkowitz, 2014*; *Chang and Martin, 2009*). In addition to the global effect of volume growth on density, the intriguing polarization of density patterns that we discovered (*Figure 6*) suggests that tip growth influences local intracellular density patterns more directly. Spatial gradients revealed that local density was correlated with tip growth, with growing ends having lower density (*Figure 6C*). It is not yet clear what cellular components are responsible for this spatial pattern, and whether they are actively depleted at growing cell tips or concentrated at non-growing regions. FITC staining revealed asymmetry in the distribution of total protein/RNA (*Figure 6—figure supplement 2*). This gradient may be caused by the distribution of membrane-bound or membrane-less organelles; it is unlikely that it arises from soluble, freely diffusing particles, unless a diffusion barrier (perhaps the nucleus) exists. We note that polarized patterns have been detected in immunofluorescence of damaged (carbonylated) proteins, which may impact replicative aging (*Erjavec et al., 2008*). Polarized density patterns established in interphase were often propagated through cell division and appeared to be inherited by the daughter cells, resulting in differences in density across a lineage (*Figure 6D,E*). How these patterns may lead to asymmetric behaviors in cell lineages such as growth patterns and aging remains to be explored (*Chang and Martin, 2009*; *Erjavec et al., 2008*). Spatial heterogeneities in density have also been observed in mammalian cells; it remains to be determined whether these patterns are related to cell shape, organelles, or variations in cytoplasmic density (*Oh et al., 2019*; *Choi et al., 2007*; *Tolde et al., 2018*).

The effects of changes in intracellular density on cellular functions are only beginning to be appreciated. The density changes of 5–20% that we observed, which likely affect the concentration of most if not all cellular components in the cytoplasm, could have profound consequences for the biochemistry of cellular reactions and on macromolecular crowding and viscosity, for instance through effects on diffusion (*Mitchison, 2019*; *Neurohr et al., 2019*; *van den Berg et al., 2017*; *Zhou et al., 2008*; *Luby-Phelps, 2000*). An intriguing implication of our findings is that the observed density variations may signify cell-cycle-dependent and spatial changes in the physical properties of the cytoplasm. Here, we provided evidence that density also affects cell mechanics and thereby influence cell shape. Intracellular osmotic pressure and density differences between daughter compartments were strongly coupled, as evidenced by bending of the elastic septal cell wall (*Figure 7*). Density may affect pressure through effects on macromolecular crowding, which produces colloid osmotic pressure associated with the displacement of water (*Mitchison, 2019*). Differences in colloid osmotic pressure have been proposed to influence nuclear size (*Mitchison, 2019*; *Harding and Feldherr, 1959*), but experimental evidence linking crowding to force generation *in vivo* generally remains scant. Our findings thus lend important support to the idea that different densities of macromolecules can generate large enough differences in colloid osmotic pressure to alter cell shape. We speculate that the increase of density at cell division may provide mechanical force through increased turgor pressure to facilitate cell-cell separation and bulging of the cell wall (*Atilgan et al., 2015*), and may affect the distribution of mechanical stresses within the cell wall during other cell-cycle stages as well.

Despite these changes in density, as a population cells maintained a relatively tight distribution of densities (CV ~6%; *Figure 1C*), suggesting the existence of homeostatic mechanisms to maintain density levels. Indeed, consistent with homeostasis, our findings indicate partial correction of density fluctuations over the course of a cell cycle (*Figure 1D*). One possible homeostasis mechanism is via regulation of volume growth rates. In fission yeast, perturbations that increase density are followed by a dramatic increase in volume growth rate that reduces density back to normal (*Knapp et al., 2019*). How density impacts volume growth rate, possibly via the concentration of certain intracellular factors or through effects on intracellular pressure, remains to be elucidated. It is also possible

that cells maintain density by tuning biosynthesis rates in other circumstances (*Miettinen et al., 2019*). Future studies focusing on the effects of cell density on particular cellular processes will be needed to understand the full scope of consequences of density variations on cell physiology.

# Materials and methods

## Key resources table

| Reagent type (species) or resource | Designation | Source or reference | Identifiers | Additional information |
|---|---|---|---|---|
| Strain, strain background (*Schizosaccharomyces pombe*) | Wild-type *S. pombe* | Other | FC15, *h⁻* wild-type strain 972 | *Figures 1*, *2*, *3*, *4*, *5* and *6*; *Figure 1—figure supplement 1*, *Figure 2—figure supplement 1*, *Figure 2—figure supplement 2*, *Figure 2—figure supplement 3*, *Figure 5—figure supplement 1*, *Figure 6—figure supplement 1* FC lab collection; https://www.uniprot.org/taxonomy/284812 |
| Genetic reagent (*S. pombe*) | *cdc25-22* mutant | Other | FC342, *h⁻cdc25-22* | *Figure 3*; FC lab collection; https://www.pombase.org/genotype/cdc25-22-C532Y-amino_acid_mutation-expression-not_assayed |
| Genetic reagent (*S. pombe*) | *cut7-446* mutant | Other | FC1455, *h⁻cut7-446 leu1-32* | *Figure 4*; FC lab collection; https://www.pombase.org/genotype/cut7-446-I954T-amino_acid_mutation-expression-knockdown |
| Genetic reagent (*S. pombe*) | *cdc16-116* mutant | Other | FC13, *h⁻cdc16-116* | *Figure 4*; FC lab collection; https://www.pombase.org/genotype/cdc16-116-unknown-unknown-expression-not_assayed |
| Genetic reagent (*S. pombe*) | *mid2* mutant | Other | FC881 *h⁻mid2::kanMX ade6 leu1-32 ura4-D18* | *Figure 7*; FC lab collection; https://www.pombase.org/genotype/mid2delta |
| Genetic reagent (*S. pombe*) | *gpd1* mutant | Other | FC3291, *h⁻gpd1::hphMX6 ade6-M216leu1-32 ura4-D18 his3-D1* | *Figure 6—figure supplement 2* FC lab collection |
| Peptide, recombinant protein | Bovine Serum Albumin | Sigma Aldrich | Cat. #: A3608 | |
| Peptide, recombinant protein | Lectin | Sigma Aldrich | Cat. #: L1395 | |
| Peptide, recombinant protein | RNAse | Thermo Scientific | Cat. #: EN0531 | |
| Chemical compound, drug | Latrunculin A | Abacam | Cat. #: ab144290 | |
| Chemical compound, drug | BODIPY 493/503 | Thermo Fisher | Cat. #: D3922 | |
| Chemical compound, drug | FITC | Sigma Aldrich | Cat. #: F7250 | |
| Chemical compound, drug | OptiPrep | Sigma Aldrich | Cat. #: D1556 | |
| Software, algorithm | Matlab | Mathworks | R2019a | |

*Continued on next page*

*Continued*

| Reagent type (species) or resource | Designation | Source or reference | Identifiers | Additional information |
|---|---|---|---|---|
| Software, algorithm | FIJI | https://imagej.net/Fiji/Downloads | v. 1.53c | |
| Software, algorithm | Algorithm to retrieve phase information | https://bitbucket.org/kchuanglab/quantitative-phase-imaging/src/master/ *Bostan et al., 2016* | | |
| Software, algorithm | Morphometrics | SimTK: Morphometrics: Project Home *Ursell et al., 2017* | | |
| Other | CellASIC Onix2 microfluidic control system | Merck | Cat. #: CAX2-S0000 | |
| Other | CellASIC ONIX microfluidic plates | Merck | Cat. #: Y04C-02-5PK | |
| Other | Optical filter | Chroma Technology | Cat. #: D680/3m | |

## Strains and cell culturing

All *S. pombe* strains used in this study are listed in the Key Resources Table. Methods for propagation and growth of *S. pombe* cells were as described in *Moreno et al., 1991*. In general, cultures were grown in 3 mL of YE5S medium at 30°C on a rotating shaker overnight to $OD_{600}$ ~1, diluted to $OD_{600}$ ~0.1, and incubated until $OD_{600}$ ~0.3 for imaging. Temperature sensitive mutant c*dc25-22* cells (and wild-type control cells) were first grown at room temperature; 90 min after imaging started, the temperature was increased to 32°C. Temperature-sensitive mutant *cdc16-116* cells (and wild-type control cells) were first grown at 25°C, then imaged on the microscope with the temperature-controlled enclosure pre-heated to 34°C. Temperature-sensitive mutant *cut7-446* cells (and wild-type control cells) were first grown at 25°C, then imaged on the microscope with the temperature-controlled enclosure pre-heated to 30°C.

## Single-cell imaging

Images were acquired with a Ti-Eclipse inverted microscope (Nikon) equipped with a 680 nm band-pass filter (D680/30, Chroma Technology) in the illumination path with a 60X (NA: 1.4) DIC oil objective (Nikon). Before imaging, Koehler illumination was configured and the peak illumination intensity at 10 ms exposure time was set to the middle of the dynamic range of the Zyla sCMOS 4.2 camera (Andor Technology). µManager v. 1.41 (*Edelstein et al., 2014*) was used to automate acquisition of z-stack bright-field images with a step size of 250 nm from ±3 µm around the focal plane (total of 25 imaging planes) to ensure substantial oversampling that facilitated correcting for potential drift in the z-direction over the course of each experiment at 5- or 10 min intervals at multiple (x,y) positions.

## Microfluidics

Cellasic microfluidic flow cell plates (Millipore, Y04C) controlled by an ONIX or ONIX2 (Millipore) microfluidic pump system were used for imaging. YE5S medium was loaded into all but one of the fluid reservoirs; the remaining well was loaded with 100 mg/mL bovine serum albumin (BSA) (Sigma Aldrich) in YE5S. Liquid was flowed from all six channels for at least 5 min at 5 psi (corresponding to 34.5 kPa), followed by 5 min of flow from YE5S-containing wells to wash out buffer and to fill channels and imaging chambers. The plate was kept in a temperature-controlled enclosure (OkoLab) throughout loading. Cells were then transferred into the appropriate well and loaded into the microfluidic imaging chamber such that a small number of cells were initially trapped, and flow of YE5S was applied. To ensure full exchange of liquid in the chamber during imaging, the flow channel was switched at least 40 s before images were acquired. Every ~2 hr, BSA flow was activated during one time point of imaging to calibrate QPI measurements.

## Image analysis to retrieve phase shifts

To reduce post-processing time, each z-stack was cropped to a square region containing the cell(s) of interest and a border of at least 40 pixels, and the focal plane was identified. This cropping was accomplished first using FIJI v. 1.53c to identify regions of interest (ROIs) within a thresholded standard deviation z-projection image of each brightfield z-stack. Using custom Matlab R2019a (Mathworks) scripts, images were cropped to the ROIs and the standard deviation of the pixels in each ROI was computed. The focal plane was defined based on the image in the stack with the lowest standard deviation. Three images above and three images below the focal plane separated by 500 nm were used to quantify cytoplasmic density. Based on these images, the phase information was calculated using a custom Matlab script implementing a previously published algorithm (*Bostan et al., 2016*). In brief, this method relates the phase information of the cell to bright-field image intensity changes along the z-direction. Equidistant, out-of-focus images above and below the focal plane are used to estimate intensity changes at various defocus distances. A phase-shift map is reconstructed in a non-linear, iterative fashion to solve the transport-of-intensity equation.

## Cytoplasmic density quantification

Using Matlab, images were background-corrected by fitting a Gaussian to the highest peak of the histogram (corresponding to the background pixels) of the phase-shift map and shifting every pixel so that the background intensity peak corresponded to zero phase shift. These background-corrected phase-shift maps were converted into binary images using watershedding for cell segmentation; where necessary, binary images were corrected manually to ensure accurate segmentation. Binary images were segmented using Morphometrics (*Ursell et al., 2017*) to generate subpixel-resolved cell outlines.

Each cell outline was skeletonized using custom Matlab code as follows. First, the closest-fitting rectangle around each cell was used to define the long axis of the cell. Perpendicular to the long axis, sectioning lines at 250 nm intervals and their intersection with the cell contour were computed. The centerline was then updated to run through the midpoint of each sectioning line between the two contour-intersection points. The slope of each sectioning line was updated to be perpendicular to the slope of the centerline around the midpoint. Sectioning lines that crossed a neighboring line were removed. Cell volume and surface area were calculated by summing the volume or area of each section, assuming rotational symmetry. Volume and area of the poles were calculated assuming a regular spherical cap.

To convert the mean intensity of the phase-shift within each cell into absolute concentration (in units of mg/mL), the mean of all cells across all time points was first calculated. Then, the decrease in phase shift induced by a prescribed concentration of BSA (typically 100 mg/mL) was defined as the difference between the mean of the phase shifts before and after the BSA imaging time point and the phase shift during the BSA time point. This difference in intensity established the calibration scaling between phase shift intensity and the concentration of BSA (*Figure 1B*). The cytoplasmic density of each cell was then calculated by dividing the mean phase shift of the cell by the aforementioned scaling factor. The mass of each cell was inferred from its mean density and volume.

## BSA calibration

Channel slides (μ-Slide VI 0.4, ibidi) were treated with lectin (Sigma-Aldrich, L1395) (0.1 mg/mL in water) for ~5 min, washed with YE5S, and cells were added and incubated for ~5 min to allow for attachment. Unattached cells were removed by washing with YE5S. Solutions of BSA (Sigma-Aldrich, A3608) in YE5S were made fresh. Attached cells in the chamber were first imaged in YE5S medium, then shifted transiently to YE5S containing different concentrations of BSA.

## Lipid droplet staining

Lipid droplets were stained with the dye BODIPY 493/503 (Thermo Fisher, D3922) (*Meyers et al., 2016*). Aliquots (10 μL) of 100 mM BODIPY in absolute ethanol were prepared. Ethanol was then evaporated in a desiccator under vacuum and dried aliquots were stored at 4°C for long-term storage. For use, an aliquot was redissolved in 10 μL absolute ethanol and 1 μL was added to 1 mL of cell culture in YE5S for each unit of cell density with $OD_{600}$ = 0.1 and incubated protected from light for ~1 min at room temperature. Cells were then pelleted at 0.4 rcf in a microfuge for 1 min and

medium was exchanged with fresh YE5S. Cells were spotted onto agarose pads and imaged with an EM-CCD camera (Hamamatsu) through a spinning-disk confocal system (Yokogawa CSU-10) attached to one of the ports of a Nikon Ti-Eclipse inverted microscope with a 488 nm laser. In parallel, bright-field z-stack images were acquired for QPI.

## Lineage tracking for time-lapse imaging datasets

First, each cell present at the beginning of the experiment was linked to the closest cell in the next frame based on the distance between centers and the difference in their size (cross-sectional area). A cell was considered the same if the centers between consecutive time points were within 20 pixels (~2 µm) and the cross-sectional area was >70% of the area at the previous time point. This process was iterated to define the lineage until either requirement was violated (usually due to cell division), at which point a new lineage was initialized using the earliest unassigned cell. All lineages were visually inspected and corrected when necessary.

## Polar growth and density quantification

To separately quantify the growth of the new and old ends, fiduciary markers such as birth scars on the cell outline were identified from which the distance to each pole at the beginning and completion of the cell cycle was measured. The density of each polar region was calculated by extracting the peak of the histogram of density values in the region within 3 µm of the pole at each time point, and then calculating the mean over time points. For FITC staining (*Knapp et al., 2019*), cells were grown in exponential phase in YE5S liquid cultures at 30℃. One milliliter of cell culture was fixed in 4% formaldehyde (Thermo Scientific, Cat. #28906) for 60 min, washed with phosphate buffered saline (PBS), and stored at 4℃. Two hundred microliters of fixed cells were then split into two separate tubes, one of which them was treated with 0.1 mg/mL RNAse (Thermo Scientific, Cat. #EN0531), and both tubes were incubated with shaking for 2 hr at 37℃. Next, cells were washed in PBS and stained with 50 ng/mL FITC (Sigma, Cat. #F7250) in PBS for 30 min, then washed three times in PBS. Cells were mounted on a PBS + 1% agarose pad and imaged in bright field and with 488 nm laser illumination on a spinning disc confocal microscope. Images were acquired with 300 nm z-steps. Monopolar cells (cells growing only at the old end) were selected manually on the basis of their cell shape, as visualized in bright-field images by a person without access to the fluorescence images. Cells were designated as monopolar based on the position of the birthscar at the new end and the characteristic curvatures of growing and non-growing cell ends. For each selected cell, FITC intensity values were calculated from images of a single medial focal plane, a sum of three medial 300 nm z-slices, or a sum of z-slices encompassing the entire cell. To measure the intensity profile along the long axis, FITC signal was measured in a stripe 1.2 µm in width along the long axis. Background intensity was subtracted, and intensities were normalized to the maximum intensity along the line profile within each cell. For comparison between the old end (OE) and new end (NE), intensities were measured in defined regions near the cell poles at 0.2–0.3 (NE) and 0.7–0.8 (OE) along the normalized cell length.

## Suspended microchannel resonator (SMR) measurements

SMR-based density and volume measurements were carried out according to a previously reported fluid-switching method (*Grover et al., 2011*). Briefly, the SMR measures the buoyant mass of a cell by flowing in culture medium through a vibrating cantilever and measuring changes in vibration frequency. The cell is then mixed with a denser medium composed of 50% culture medium and 50% OptiPrep (Sigma-Aldrich), and flowed back through the cantilever to obtain a second buoyant mass measurement 10 s later. Cell volume and density are calculated from the two consecutive buoyant mass measurements based on the known densities of the two fluids (*Grover et al., 2011*). Each cell is serially flushed into the system in culture medium from a reservoir at 30℃. After every hour of measurement, the reservoir is replenished from an exponential-phase culture.

SMR devices were fabricated at CEA-LETI (Grenoble, France). The physical dimensions and operation of the SMR, as well as data analyses, were identical to those reported in *Miettinen et al., 2019*; *Kang et al., 2019*. Briefly, the SMR cantilever was vibrated in the second flexural bending mode using a piezo-ceramic plate underneath the SMR chip. The vibration frequency of the cantilever was measured using piezo-resistors at the base of the cantilever. A digital control platform was used to

drive the cantilever in a feedback mode, where the vibration frequency signal acquired from the piezo-resistor was delayed, amplified, and used as the drive signal to actuate the cantilever. Fluid flow was controlled using two electronic pressure regulators and solenoid valves, which were used to pressurize vials containing the culture medium. A typical cell transit time through the cantilever was 150 ms. System temperature was controlled by mounting the SMR and culture-medium vials on copper stands connected to a heated water bath. All SMR operations were controlled using National Instruments control cards and custom LabVIEW (2012) code. Frequency data were analyzed using previously reported custom Matlab code that measures the maximum frequency change during the transit of each cell through the cantilever (*Kang et al., 2019*). Frequency measurements were calibrated using polystyrene beads and NaCl solutions of known density.

## Holographic refractive index measurements

For refractive index measurements, wild-type *S. pombe* cells grown at 30°C were immobilized on a lectin-coated glass-bottom 35 mm diameter μ-dish (ibidi). Holographic refractive index measurements were acquired with a 3D Cell Explorer system (Nanolive) with a temperature-controlled enclosure set to 30°C. First, sum images of *z*-stacks of three-dimensional refractive index maps were generated to retrieve cell outlines by watershedding. Cells oriented at an angle to the flat glass bottom dish were ignored. For each remaining cell, the mean refractive index was extracted from each image in the *z*-stack using Matlab and the highest value (assumed to correspond to the middle plane) was used for further analysis.

## Latrunculin A treatment

Stock solutions were made by dissolving 100 μg latrunculin A (Abacam, ab144290) in dimethyl sulfoxide (DMSO, Sigma-Aldrich) to a concentration of 20 mM and stored at −20°C in 1 μL aliquots. To prepare agarose pads, 1 μL of 20 mM latrunculin A or 1 μL of DMSO was mixed with 100 μL of YE5S medium containing 2% (w/v) agarose UltraPure agarose (Invitrogen Corporation) kept in a water bath at ~70°C. The mixture was pipetted onto a microscope glass slide and quickly covered with another slide to form flat agarose pads with thickness of ~2 mm. Once pads had solidified, one slide was carefully removed and 1–2 μL of exponential-phase wild-type cells were deposited on the agarose pad. Cells were allowed to settle for 1–2 min before a coverslip was placed on top and sides were sealed with Valap (1:1:1 vaseline:lanolin:paraffin) to prevent evaporation during imaging.

## Statistical analyses

The magnitude of the correlation between two continuous variables was reported using Pearson's correlation coefficient. $R^2$ and associated p-values were calculated using the built-in function corrcoef in Matlab R2019a (Mathworks). Experiments are representative of at least two biological replicates with independent data sets.

# Acknowledgements

We thank the Chang and Huang labs for discussion and support, Gabriella Estevam for her contributions to the project at its early stages, Scott M Knudsen for assisting with cell culturing for SMR measurements, and Sophie Dumont and her lab for discussion. PDO was supported by postdoctoral fellowships from the Swiss National Science Foundation under Grants P2ELP3_172318 and P400PB_180872. FC was supported by NIH GM056836. TPM received funding from the Wellcome Trust (110275/Z/15/Z). KCH is a Chan Zuckerberg Biohub investigator.

# Additional information

### Funding

| Funder | Grant reference number | Author |
|---|---|---|
| Swiss National Science Foundation | P2ELP3_172318 | Pascal D Odermatt |
| Swiss National Science Foundation | P400PB_180872 | Pascal D Odermatt |

| National Institute of General Medical Sciences | NIH GM056836 | Fred Chang |
|---|---|---|
| Wellcome Trust | 110275/Z/15/Z | Teemu P Miettinen |
| Chan Zuckerberg Initiative | | Kerwyn Casey Huang |

The funders had no role in study design, data collection and interpretation, or the decision to submit the work for publication.

## Author contributions
Pascal D Odermatt, Conceptualization, Resources, Data curation, Software, Formal analysis, Funding acquisition, Validation, Investigation, Visualization, Methodology, Writing - original draft, Writing - review and editing; Teemu P Miettinen, Investigation, Methodology, Writing - review and editing; Joël Lemière, Validation, Investigation, Visualization, Methodology; Joon Ho Kang, Investigation, Methodology; Emrah Bostan, Conceptualization, Software, Formal analysis, Investigation, Methodology, Writing - review and editing; Scott R Manalis, Supervision, Funding acquisition, Methodology, Writing - review and editing; Kerwyn Casey Huang, Conceptualization, Resources, Data curation, Supervision, Funding acquisition, Methodology, Writing - original draft, Project administration, Writing - review and editing; Fred Chang, Conceptualization, Supervision, Funding acquisition, Methodology, Writing - original draft, Project administration, Writing - review and editing

## Author ORCIDs
Pascal D Odermatt (iD) https://orcid.org/0000-0002-1307-1364
Teemu P Miettinen (iD) https://orcid.org/0000-0002-5975-200X
Joël Lemière (iD) https://orcid.org/0000-0002-9017-1959
Joon Ho Kang (iD) http://orcid.org/0000-0003-4165-7538
Kerwyn Casey Huang (iD) https://orcid.org/0000-0002-8043-8138
Fred Chang (iD) https://orcid.org/0000-0002-8907-3286

## Decision letter and Author response
Decision letter https://doi.org/10.7554/eLife.64901.sa1
Author response https://doi.org/10.7554/eLife.64901.sa2

# Additional files

## Supplementary files
- Transparent reporting form

## Data availability
All data generated or analysed during this study are included in the manuscript and supporting files. Custom Matlab code used for image analysis has been posted online at https://bitbucket.org/kchuanglab/quantitative-phase-imaging/src/master/ (copy archived at https://archive.softwareheritage.org/swh:1:dir:789a30703e749c6c456c6e7097878f743b2007ba).

The following dataset was generated:

| Author(s) | Year | Dataset title | Dataset URL | Database and Identifier |
|---|---|---|---|---|
| Odermatt PD | 2021 | Code for Quantitative Phase Imaging | https://bitbucket.org/kchuanglab/quantitative-phase-imaging/src/master/ | Matlab code to computationally retrieve quantitative phase information from a stack of brightfield images, quantitative-phase-imaging |

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
