## [Decision Letter]

**Acceptance summary:**

This article contributes to the fundamental understanding of how a cell grows. It provides a broadly applicable method for dry mass measurement of single cells and, using it, it describe how cell density varies across the cell division cycle. The key finding of this article is the fact that growth in mass and volume seem to be generally uncoupled, leading to significant density changes. The method developed can be used across multiple biological systems to investigate similar questions.

**Decision letter after peer review:**

Thank you for submitting your article "Variations of intracellular density during the cell cycle arise from tip-growth regulation in fission yeast" for consideration by *eLife*. Your article has been reviewed by 3 peer reviewers, and the evaluation has been overseen by Mohan Balasubramanian as the Reviewing Editor and Naama Barkai as the Senior Editor. The following individual involved in review of your submission has agreed to reveal their identity: Matthieu Piel (Reviewer #3).

The editors and referees agreed that you have developed a novel and highly quantitative approach to investigate some key questions in cell physiology that require precise measurements of cell dry mass density and how this tracks cell cycle progression. The referees also paid particular attention to the density gradients that you have identified, which all agreed was exciting. However, a small number of issues were raised both in the review process and also during further discussion.

I think most issues raised can be fixed with appropriate rewriting, although you may need new data in 1-2 cases.

Below I have summarized the essential revisions required for further consideration. The referee’s comments also are appended below verbatim.

Essential revisions:

1. Density gradients (5-10%) are apparent in G2 cells, with lower density at faster growing (old end) poles. The gradients persist through mitosis and cytokinesis, giving birth to daughter cells whose mean densities reliably differ from each other. These findings would appear to suggest that density differences might accumulate with each generation, but this is clearly not the case as the density variation in a cell population is very small (CV ~ 6%). This raises the interesting question of how density homeostasis is maintained, which this paper does not address but could clearly be investigated with available tools/mutants/conditions.

2. Supp Figure 4: The authors' surface area measurements require estimation of septum surface area. Those estimates assume a relation between optical density and fractional septum diameter (rather than fractional septal area?) that would seem to need validation.

3. To me, the most interesting result is that density gradients a stable within cells. This result must have important implications in cytosolic viscosity, which I was disappointed was not explicitly discussed. The discussion does claim that the "distributions of large organelles and total protein are not polarized", but it is not clear that the papers cited to support that claim would be able to detect the ~5% reported difference in density. The organelle paper contains no quantitation and the noise in the protein paper looks to be around 10%.

4. I was surprised that the density increases in the control cells in Figure 5B. That result seems to contradict the result that density decreases during interphase (Figure 1C). The authors should address this apparent contradiction.

5. I could not understand how it is possible that internal density gradients are established. I can see two options, which the authors might want to discuss: a) a part of what they measure in dry mass is dominated by the mass of the cell wall, especially near the cell tips. In this case, faster growth of the tip could be associated with a lower density (or thinner) wall, if the same amount of material is added per unit of time (which could be the case if that rate is given by a rate of transport, while elongation rate is related to the turgor pressure and the cell wall mechanics and thus independent). b) Cell dry mass is dominated by the assembly of a solid like structure (maybe a sort of gel? sugars or large polymers?) which would be less dense when it assembles faster at the faster elongated tip. A bit the same as the other hypothesis, but not restricted to the cell wall, rather a cytoplasmic gel like structure.

6. The second strange observation is the relation between dry mass density and the bending of the septum. There are two interesting aspects: where the pressure would come from? It is not clear that the type of component which usually dominates dry mass (and thus density here), usually proteins or other large biomolecules, would the one dominating the turgor pressure, usually produced by more numerous but smaller solutes (ions, small sugars like glycerol). What could give this correlation? Charges of large molecules? Or could it be, combining with the point 1) above, that a sort of gel is formed during growth, which dominates dry mass, and would also entrap solutes which produce the largest turgor pressure?

It is a quite important point, as it could be a fundamental mechanism coupling volume and dry mass.

7. The authors already discuss a little about density homeostasis, but it seems that there should be more emphasis given to the fact that some uncoupled growth of dry mass and volume could give rise to such a narrow distribution of density. They did not detect any negative correlation between density and mass growth rate, which could correct density on long timescales? Maybe due to molecular crowding?

*Reviewer #1 (Recommendations for the authors (required)):*

This paper uses an optical measure of dry mass density that appears easier to use than several other strategies to investigate how density varies as a function of the cell cycle in fission yeast. They find that density decreases during G2 and increases during mitosis/cytokinesis. The idea that dry mass growth continues while volume growth stops during mitosis, leading to cell cycle fluctuations in density, dates back to the early work of JM Mitchison. Data on cdc mutants and Latrunculin treatment are consistent with the idea that while mass increases continuously, cell cycle-regulated changes in volume growth create the density oscillations, which seems unsurprising. One difference between the current work and older work is that JM Mitchison thought the dry mass increase was linear, while here it is argued to be exponential (consistent with some other work on protein synthesis). Overall, the quantitative data appeared to me to convincingly support the authors' conclusions.

Perhaps the more interesting findings come in the last two figures. Density gradients (5-10%) are apparent in G2 cells, with lower density at faster growing (old end) poles. The gradients persist through mitosis and cytokinesis, giving birth to daughter cells whose mean densities reliably differ from each other. These findings would appear to suggest that density differences might accumulate with each generation, but this is clearly not the case as the density variation in a cell population is very small (CV ~ 6%). This raises the interesting question of how density homeostasis is maintained, which this paper does not address but could clearly be investigated with available tools/mutants/conditions.

Technical point:

Supp Figure 4: The authors' surface area measurements require estimation of septum surface area. Those estimates assume a relation between optical density and fractional septum diameter (rather than fractional septal area?) that would seem to need validation.

*Reviewer #2 (Recommendations for the authors (required)):*

I was surprised that the density increases in the control cells in Figure 5B. That result seems to contradict the result that density decreases during interphase (Figure 1C). The authors should address this apparent contradiction.

*Reviewer #3 (Recommendations for the authors (required)):*

1. I could not understand how it is possible that internal density gradients are established. I can see two options, which the authors might want to discuss: a) a part of what they measure in dry mass is dominated by the mass of the cell wall, especially near the cell tips. In this case, faster growth of the tip could be associated with a lower density (or thinner) wall, if the same amount of material is added per unit of time (which could be the case if that rate is given by a rate of transport, while elongation rate is related to the turgor pressure and the cell wall mechanics and thus independent). b) Cell dry mass is dominated by the assembly of a solid like structure (maybe a sort of gel? sugars or large polymers?) which would be less dense when it assembles faster at the faster elongated tip. A bit the same as the other hypothesis, but not restricted to the cell wall, rather a cytoplasmic gel like structure.

2. The second strange observation is the relation between dry mass density and the bending of the septum. There are two interesting aspects: where the pressure would come from? It is not clear that the type of component which usually dominates dry mass (and thus density here), usually proteins or other large biomolecules, would the one dominating the turgor pressure, usually produced by more numerous but smaller solutes (ions, small sugars like glycerol). What could give this correlation? Charges of large molecules? Or could it be, combining with the point 1) above, that a sort of gel is formed during growth, which dominates dry mass, and would also entrap solutes which produce the largest turgor pressure?

It is a quite important point, as it could be a fundamental mechanism coupling volume and dry mass.

3. The authors already discuss a little about density homeostasis, but it seems that there should be more emphasis given to the fact that some uncoupled growth of dry mass and volume could give rise to such a narrow distribution of density. They did not detect any negative correlation between density and mass growth rate, which could correct density on long timescales? Maybe due to molecular crowding?

---

## [Author Response]

Essential revisions:1. Density gradients (5-10%) are apparent in G2 cells, with lower density at faster growing (old end) poles. The gradients persist through mitosis and cytokinesis, giving birth to daughter cells whose mean densities reliably differ from each other. These findings would appear to suggest that density differences might accumulate with each generation, but this is clearly not the case as the density variation in a cell population is very small (CV ~ 6%). This raises the interesting question of how density homeostasis is maintained, which this paper does not address but could clearly be investigated with available tools/mutants/conditions.

We thank the reviewers for this excellent suggestion. We have now added important new data demonstrating density homeostasis (new Figure 1D). Similar to cell size homeostasis analyses, we plotted the change in density over the entire cell cycle as a function of the density at the beginning of the cell cycle. The plot revealed an inverse relationship consistent with homeostatic behavior over the cell cycle, whereby cells with higher or lower than average density at the beginning of the cell cycle end up closer to the average by the end of the cell cycle.

2. Supp Figure 4: The authors' surface area measurements require estimation of septum surface area. Those estimates assume a relation between optical density and fractional septum diameter (rather than fractional septal area?) that would seem to need validation.

As suggested, we changed our measurements to use the intensity of the septa in QPI images to estimate septal surface area instead of diameter. We note that we used the intensity of the QPI signal because it was not possible to accurately measure the dimensions of the septa directly from the QPI images.

The new results are shown in Figure 2-supplement 3 and show that the ratio of surface area to mass remains relatively constant throughout the cell cycle. We have modified the discussion of this figure accordingly, these changes do not affect the general conclusions of our study.

3. To me, the most interesting result is that density gradients a stable within cells. This result must have important implications in cytosolic viscosity, which I was disappointed was not explicitly discussed. The discussion does claim that the "distributions of large organelles and total protein are not polarized", but it is not clear that the papers cited to support that claim would be able to detect the ~5% reported difference in density. The organelle paper contains no quantitation and the noise in the protein paper looks to be around 10%.

In response to this comment, we now provide new data on the spatial distribution of intracellular contents that support the observed gradient in density. We show that FITC staining of total protein and RNA reveals similar spatial gradients as for density, in which the intensity at the old end is on average 5% less than at the new end in cells growing only at one pole. Similar differences between the cell ends were observed in analyses of FITC intensity data from single or multiple focal planes. These data are now included in new Figure 6-supplement 2. These findings strongly support the existence of an intracellular density gradient.

We agree that currently available literature are not able to conclusively determine whether organelle distribution contributes to this gradient distribution and hence have revised our point regarding organelle distribution.

We have included in the discussion the exciting possibility that these density changes may affect spatial and temporal variation in crowding, diffusion, and viscosity in the cytoplasm.

4. I was surprised that the density increases in the control cells in Figure 5B. That result seems to contradict the result that density decreases during interphase (Figure 1C). The authors should address this apparent contradiction.

The apparent density rise in the control cells over time is due to the nature of the measurement method and the lack of synchronization among the population at the time of treatment. As each cell completed a cell cycle and divides, it was thereafter no longer counted in the cohort. Since the cells were asynchronous, over time the tracked population was enriched for cells in mitosis or cell division, which experience an increase in density (Figure 1C). We have clarified this point in the text.

5. I could not understand how it is possible that internal density gradients are established. I can see two options, which the authors might want to discuss: a) a part of what they measure in dry mass is dominated by the mass of the cell wall, especially near the cell tips. In this case, faster growth of the tip could be associated with a lower density (or thinner) wall, if the same amount of material is added per unit of time (which could be the case if that rate is given by a rate of transport, while elongation rate is related to the turgor pressure and the cell wall mechanics and thus independent). b) Cell dry mass is dominated by the assembly of a solid like structure (maybe a sort of gel? sugars or large polymers?) which would be less dense when it assembles faster at the faster elongated tip. A bit the same as the other hypothesis, but not restricted to the cell wall, rather a cytoplasmic gel like structure.

Similar to the reviewer, we were surprised to observe the intracellular density gradients. To validate these findings, we have added important new data using FITC staining showing that total protein and RNA also exhibit a gradient (new Figure 6-supplement 2). As FITC does not stain the cell wall, these data argue against the gradient being due to variations in the mass of the cell wall. Moreover, we note that our QPI method may not recognize the lateral cell wall; for instance, birth scars, which are regions of thicker cell wall, do not exhibit higher intensity by QPI.

We have added speculation to the text that this stable spatial density gradient might be due to asymmetric distributions of organelles or other structures that do not diffuse freely. We hope to study the basis of this gradient further in future studies.

6. The second strange observation is the relation between dry mass density and the bending of the septum. There are two interesting aspects: where the pressure would come from? It is not clear that the type of component which usually dominates dry mass (and thus density here), usually proteins or other large biomolecules, would the one dominating the turgor pressure, usually produced by more numerous but smaller solutes (ions, small sugars like glycerol). What could give this correlation? Charges of large molecules? Or could it be, combining with the point 1) above, that a sort of gel is formed during growth, which dominates dry mass, and would also entrap solutes which produce the largest turgor pressure?It is a quite important point, as it could be a fundamental mechanism coupling volume and dry mass.

We agree, and have emphasized this point about the coupling between volume and dry mass in the discussion. We propose that macromolecules exert colloid osmotic pressure, as discussed in Mitchison’s MBoC review. This colloid pressure may be smaller than the turgor pressure produced by small solutes such as ions, but may cause enough difference between the compartments to bend the septum. We agree that how density regulates volume growth rate is a very important question that we plan to address in the future.

7. The authors already discuss a little about density homeostasis, but it seems that there should be more emphasis given to the fact that some uncoupled growth of dry mass and volume could give rise to such a narrow distribution of density. They did not detect any negative correlation between density and mass growth rate, which could correct density on long timescales? Maybe due to molecular crowding?

We have now provided new data regarding density homeostasis (Figure 1D) and added text to the discussion about potential mechanisms by which this homeostasis could be achieved. We intend to investigate this homeostatic mechanism in the future.